

# Beneficial dose-dependent effects of Ag nanoparticles on germination do not compromise growth and metabolic profiles of *Capsicum annuum* seedlings

Berenice Cortes Espinoza[*], Alejandro Sánchez-González, María Evarista Arellano-García and Rafael Bello-Bedoy[*]

Facultad de Ciencias, Universidad Autónoma de Baja California, Ensenada, Baja California, Mexico
[*] These authors contributed equally to this work.

Corresponding author
Rafael Bello-Bedoy,
rbello@uabc.edu.mx

## ABSTRACT

This study evaluates the effects of silver nanoparticles (AgNPs) as a nanopriming agent and their potential detrimental impacts on growth and physiology in wild and domesticated chili (*Capsicum annuum*) seeds. We compared the responses of wild (*C. annuum* var. *glabriusculum*) and domesticated (Serrano) plants. Seeds were soaked for 24 hours in AgNP solutions at concentrations of 0 ppm, 50 ppm, 100 ppm, and 250 ppm. Germination was monitored daily over a 14-day period in replicated Petri dishes under controlled growth chamber conditions. A subsample of germinated seedlings from each treatment was transplanted into plastic pots to assess plant growth and secondary metabolism at 28 and 42 days after germination. On each sampling day, three randomly selected plants per treatment were evaluated for shoot and root length, as well as wet and dry biomass. Physiological measurements included both primary and secondary metabolites, specifically chlorophyll and polyphenols. Potential genotoxic effects were assessed by exposing meristematic root tissue to a 5 ppm AgNP solution for 72 hours and analyzing mitotic activity. The results showed that AgNPs significantly increased germination in wild chili, reaching 90% compared to 77% in the control, without negative effects on plant development. No significant differences were observed in growth traits or polyphenol content, or the number of dividing cells. Furthermore, no chromosomal aberrations were detected. The beneficial effects of nanopriming were limited to the germination stage in wild plants, and no enhancement was observed in the domesticated variety. These results suggest that domestication may reduce sensitivity to nanopriming. Overall, our findings support the potential benefits and safety of AgNP-based nanopriming in agriculture, even at high concentrations.

## INTRODUCTION

The germination process is critical for successful plant establishment, which is essential for the conservation of wild populations and the efficient production of resources in domesticated plants (*Hay & Probert, 2013*). This is particularly relevant for high-value

crops that include both wild and domesticated varieties, where there is a need to conserve the genetic resources of wild relative to enhance yield (*Mastretta-Yanes et al., 2024*) and to preserve biological diversity and cultural heritage (*Tobón-Niedfeldt et al., 2022*). The adoption of technologies that support both wild plant conservation and crop improvement is therefore highly desirable. In recent years, nanoparticles have emerged as a promising tool to address challenges in agriculture (*Mahakham et al., 2017*; *Song & He, 2021*; *Singh et al., 2023*), and their beneficial effects may be applicable to wild plant species. However, the use of nanoparticles can also lead to negative impacts on plant growth and reproduction, including genotoxicity and phytotoxicity. Therefore, to address the problem of low germination and establishment due to seedling sensitivity to nanoparticle-induced stress, it is necessary to evaluate the potential and safety of this technology.

Seed priming with nanoparticles (*i.e.*, nanopriming) is a pre-sowing treatment that involves the use of various types of nanoparticles suspended in an aqueous solution (*Nile et al., 2022*). Nanopriming can alter physiological and biochemical processes, positively influencing germination, growth, and metabolite synthesis in several crops (*Mahakham et al., 2017*; *Acharya et al., 2020*; *Almutairi & Alharbi, 2015*; *Imtiaz et al., 2023*). However, it has also been associated with adverse effects on genetic material and plant development, including chromosomal aberrations, nuclear breaks, and reduced growth such as inhibited root elongation (*i.e.,* genotoxicity and phytotoxicity) (*Kumari, Mukherjee & Chandrasekaran, 2009*; *Patlolla et al., 2012*; *Thuesombat et al., 2014*; *Almutairi & Alharbi, 2015*; *De Paiva Pinheiro et al., 2020*). These negative impacts raise concerns about the efficacy and safety of nanoparticle use. This is particularly relevant in scenarios where nanoparticles may accumulate in soil and water, potentially becoming an environmental issue.

The effects of nanopriming with silver nanoparticles (AgNPs) in plant biology are complex. Recent reviews of empirical evidence indicate that plant responses vary significantly depending on genetic background and AgNP doses (*Imtiaz et al., 2023*; *Khan et al., 2023*). For instance, the germination response of diploid and triploid watermelon varieties differed: nanopriming enhanced germination and early vegetative growth in the diploid variety, but had no effect on the triploid (*Acharya et al., 2020*), suggesting that the effects of silver nanoparticles depend on genetic composition. Similarly, the concentration of AgNPs used in nanopriming treatments contributes to variation in outcomes (*Imtiaz et al., 2023*). Several studies have reported dose-dependent effects of AgNP nanopriming on germination, early growth, and development in crops such as wheat, rice, watermelon, and zucchini (*Almutairi & Alharbi, 2015*; *Acharya et al., 2020*; *Santhoshkumar, Hima Parvathy & Soniya, 2024*). In contrast, negative effects have been observed at higher concentrations in onion and wheat (*Kumari, Mukherjee & Chandrasekaran, 2009*; *Vannini et al., 2014*). Overall, the effects of AgNP nanopriming and dosage appear to be highly species-dependent.

One of the main sources of genetic differentiation in plants is artificial selection, particularly through domestication and genetic improvement processes (*Mostert-O'Neill et al., 2022*). The resulting divergence between domesticated varieties and their wild progenitors can alter plant responses to exogenous biotic and abiotic factors (*Milla et al., 2015*), as plant domestication typically alter germination, growth requirements, and the

synthesis of secondary metabolites (*Shlichta et al., 2018*; *Munguía-Rosas, 2021*; *Serrano-Mejía et al., 2022*). Since domestication modifies various plant traits, it is reasonable to expect that wild and domesticated plants respond differently to nanoparticle exposure. Based on this evidence, it is expected to find variation in the response on germination and growth in wild and domesticated plants exposed to silver nanoparticles.

Mexican *Capsicum annuum* includes the wild relative *C. annuum var. glabriusculum* and more than 60 domesticated varieties that have been subject to selection and breeding by pre-Hispanic cultures and generations of farmers over approximately 6,400 years in Mexico (*Araceli et al., 2009*; *Kraft et al., 2014*). As a result of domestication, wild and domesticated chili plants exhibit marked phenotypic differences. *Serrano Mejía (2023)* documented substantial divergence in plant architecture, as well as significant increases in leaf, fruit, and seed size in domesticated varieties, traits that influence reproduction, seedling establishment, and agronomic performance. For instance, seeds of wild *C. annuum* var. *glabriusculum* typically show low germination rates (*Hernández-Verdugo et al., 2010*; *Cano-Vazquez et al., 2015*), which has been linked to their smaller size and limited nutrient reserves, both of which affect germination and early growth (*García Federico et al., 2010*). In contrast, domesticated plants have undergone strong selection for improved fruit traits, which has indirectly favored the development of larger seeds. This suggests that domesticated varieties may have higher germination and seedling growth potential. However, no published studies have reported germination rates or establishment success in domesticated Mexican chili varieties. In the case of wild populations, several studies have examined germination, but no consistent method has been identified to improve germination rates (*Hernández-Verdugo, Oyama & Vázquez-Yanes, 2001*; *Granata et al., 2024*). Investigating innovative strategies to improve seed germination and seedling growth is essential for maximizing the fitness of wild relatives and improving crop yield.

This study investigates the effects of silver nanoparticles (AgNPs) on germination, seedling growth, and morphology in wild and domesticated chili plants *Capsicum annuum*, with particular attention to dose-dependent responses. Our objective is to identify concentrations that enhance early plant performance while also evaluating potential phytotoxic and genotoxic risks. Germination and seedling growth were assessed at three AgNP concentrations. In parallel, genotoxic effects were evaluated separately through analysis of the root mitotic index (*Sánchez-Pérez et al., 2023*). This dual approach allows us to determine whether AgNPs enhance early development or induce phytotoxic or genotoxic responses, and whether these effects differ between wild and domesticated genetic backgrounds. We hypothesize that AgNPs will have a greater positive effect on wild plants, which typically possess smaller seeds and exhibit greater variability in germination and growth, whereas domesticated plants—with larger seed reserves and consistently high performance—will show limited or no additional benefit.

## MATERIALS & METHODS

### Study species

*Capsicum annuum* L. (Solanaceae) is a perennial, flowering shrub that comprises both the wild populations (*Capsicum annuum* var. *glabriusculum*) and domesticated varieties

(*Capsicum annuum* var. *annuum*). The height of wild and domesticated plants typically ranges from 50 cm to 1.5 m (*Solís-Montero, Bello-Bedoy & Munguía-Rosas, 2023*). The leaves of wild plants are generally smaller and differ in shape compared to those of domesticated varieties (*Serrano-Mejía et al., 2022*). Both wild and domesticated plants develop a taproot system with branched secondary roots. Fruit shape, size, varies widely among varieties. The flowers are typically small and white, with five petals (*Serrano Mejía, 2023*).

## Seed germination trials

Germination trials were conducted using seeds from the domesticated chili variety *Capsicum annuum* var. *annuum* (Serrano) and its wild counterpart, *Capsicum annuum* var. *glabriusculum*. Wild chili seeds were collected from natural populations in Sonora, Mexico, while Serrano seeds were obtained from a commercial supplier. The average seed weight for the Serrano variety was 4.16 mg (207.9 mg/50 seeds), whereas the wild chili seeds averaged 2.6 mg (132.7 mg/50 seeds).

Prior to the germination assays, seeds were surface-sterilized with 70% ethanol for 2 min and rinsed three times with distilled water. Ten seeds of each variety were placed in eight cm diameter Petri dishes lined with filter paper, which provided uniform support, ensured even moisture distribution, and prevented waterlogging during germination. Each treatment was replicated five times. Seeds were treated with five mL of one of four AgNP solutions: 0 (control), 50, 100, or 250 ppm.

Each treatment was replicated four times. The Petri dishes were sealed with Parafilm to prevent moisture loss and contamination, and then placed in a controlled-environment growth chamber maintained at 24–27 °C, with a 16:8 h light/dark photoperiod and 65% relative humidity. Seeds were incubated under these conditions for 14 days.

The silver nanoparticles used are part of a commercially available veterinary formulation previously characterized in detail (see *Bello-Bello et al., 2018* for a detailed description on Argovit™). These nanoparticles exhibit spheroidal morphology, a size range of 1–90 nm (mean ∼35 nm), polyvinylpyrrolidone (PVP) stabilization, and a silver content of 1.2% (w/w), as confirmed by TEM, DLS, FTIR, and UV-Vis spectroscopy (*Bello-Bello et al., 2018*; *Stephano-Hornedo et al., 2020*). In this study, a concentrated stock solution was used to prepare the experimental dilutions, ensuring precise control of the final concentrations and facilitating reproducibility and comparability with previous studies.

## Germination percentage

A seed was registered as germinated when radicle emergence was observed. Germination was recorded in a binomial format: 1 = germinated, 0 = not germinated. Germination was monitored daily for 14 days following treatment, corresponding to the period during which most seeds completed germination. We calculated both total germination percentage and germination rate. Total germination percentage was defined as the ratio of germinated seeds to the total number of seeds incubated.

At germination, each seed was transplanted individually into 50-cell trays filled with sterile commercial substrate (Berger BM2) for initial growth. Once seedlings developed two

fully expanded leaves, they were individually transplanted into 3.5-inch pots containing the same substrate. Subsequently, when seedlings reached four leaves, they were transferred to 2.5 L pots to allow for continued growth. Wild and domesticated chili plants develop extensive root systems; thus, seedlings were transplanted twice during the experiment to ensure adequate root space and minimize restriction. All transplanting procedures were carried out carefully to avoid root damage and reduce transplant stress. Plants were irrigated with 100 mL of distilled water every other day to maintain consistent soil moisture. Additionally, seedlings received a weekly application of 25 mL of a 3 g L$^{-1}$ NPK (19-19-19) nutrient solution.

## Growth measurements

To assess the effects of nanopriming on the growth and development of wild and domesticated chili plants, the following traits were measured at 28 and 42 days after germination: root length, shoot length, plant height, and wet and dry biomass of both root and shoot. Plants were cultivated in a controlled-environment chamber set to 24–27 °C, with a 16:8 h light/dark photoperiod and 65% relative humidity.

Measurements of plant growth plant length and biomass were carried out as follow: Root length was measured from the base of the stem to the tip of the primary root. Shoot length was measured from the stem base to the apical bud of the main stem. Wet biomass was measured by carefully removing the roots from the substrate, rinsing them thoroughly with distilled water, drying them with blotting paper, and weighing the plant material using a precision balance (Mettler Toledo MSQ205) to the nearest 0.0001 g. For dry biomass, the samples were wrapped in aluminum foil and dehydrated in a VWR convection oven at 65 °C for seven days to ensure complete desiccation. The dried material was then weighed using the same precision balance.

## Total polyphenol content

To quantify total polyphenol content, fresh plant leaves were analyzed at 28 and 42 days after germination using the Folin–Ciocalteu method. First, properly labeled Eppendorf tubes were weighed. Leaves were then cleaned with moistened blotting paper to remove any substrate residue. Once cleaned, each sample was individually macerated in an Eppendorf tube and weighed. Samples were then incubated with 1 mL of methanol–water solution (80:20 v/v) for 24 h in the dark to prevent light-induced degradation and oxidation of phenolic compounds, thereby preserving their chemical stability during extraction.

After 24 h, the samples were centrifuged at 13,000 rpm for 5 min. In new Eppendorf tubes, 1 mL of distilled water, 20 µL of extract from each sample, and 100 µL of Folin–Ciocalteu phenol reagent (Sigma-Aldrich, St. Louis, MO, USA) were added and mixed thoroughly. Subsequently, 300 µL of 20% (w/v) sodium carbonate ($Na_2CO_3$) solution was added and mixed. The reaction mixtures were incubated at room temperature for 2 h. After incubation, the absorbance of each sample was measured at 765 nm using a Hach DR 2800 spectrophotometer. Gallic acid was used as the standard for constructing a calibration curve with standard concentrations of 50, 100, 150, 250, 500, 1,000, 1,500, and 2,000 mg/L, following the protocol of *Ainsworth & Gillespie (2007)*. The regression

equation derived from the standard curve was used to calculate the gallic acid equivalent (GAE) concentration in each sample. Total polyphenol content was expressed as milligrams of GAE per gram of fresh leaf tissue. Distilled water was used as the blank control.

## Chlorophyll content

To quantify chlorophyll content, the leaf chlorophyll content index (CCI) was measured using a Chlorophyll Content Meter-200 Plus (OPTI-SCIENCE, Hudson, NH, USA). This device estimates chlorophyll concentration by measuring light absorbance at 600 nm and 900 nm ($\pm 1.0$ nm) over a leaf area of 0.71 cm$^2$. Measurements were taken from the fourth fully expanded true leaf of each plant. For each leaf, three readings were obtained and their average was used as the final CCI value.

## Cell division indicators

To evaluate the potential cytotoxic effects of nanopriming on cell division in apical zones, roots and shoots were exposed to an AgNP solution, with water used as the control. Seeds from both wild and domesticated *Capsicum annuum* were germinated, and once seedlings developed three to five true leaves, 20 individuals from each variety were randomly selected for the experiment. Five plants were assigned to each treatment group. The seedlings were carefully uprooted, rinsed, and placed in 150 mL beakers containing 120 mL of either the AgNP solution (5 ppm) or distilled water. Plants remained in the treatments for 72 h. Shoot and root lengths were measured every 24 h using a ruler, and digital images were analyzed with ImageJ software.

After 72 h, root samples were collected for cytological staining to visualize dividing cells. To examine cell division and calculate mitotic and phase indices. To accomplish this, root tips were prepared following a modified version of the cell division observation protocol described by *Rohami et al. (2010)*. Briefly, lateral roots were excised from the apical zone at approximately 1 cm from the tip. The root sections were individually placed in 1.5 mL microcentrifuge tubes containing a fixative solution of ethanol and glacial acetic acid (3:1, v/v) for 24 hours at room temperature. Subsequently, hydrolysis was performed with 1 N hydrochloric acid (HCl) for 30 minutes at room temperature on a watch glass. The roots were then individually stained on microscope slides using Aceto-Orcein (Sigma-Aldrich, St. Louis, MO, USA) for 30 minutes at room temperature. After staining, the roots were rinsed twice with acetic acid to remove the excess dye. Finally, each root tip was mounted on a microscope slide, treated with a drop of acetic acid, and gently squashed using a pencil eraser to evenly spread the cells.

Stained root tip samples were examined under a light microscope at 100× magnification using immersion oil. For each sample, between 500 and 2,000 cells were counted to calculate the mitotic index and determine the distribution of cells across the different mitotic phases. The mitotic index was calculated as the ratio of dividing cells to the total number of observed cells (*Howell et al., 2007*). The phase index was determined by calculating the proportion of cells in each mitotic phase—prophase, metaphase, anaphase, and telophase—relative to the total number of cells observed.

**Table 1  Germination success of *Capsicum annuum* varieties (wild *vs.* cultivated), treatment of silver nanoparticles exposure, and its interaction.** Estimates of the logistic regression of germination success as a function of *Capsicum annuum* variety (wild *vs.* cultivated), treatment of silver nanoparticles exposure, and its interaction.

| Source of variation | d.f. | $X_2L$-R | P value |
|---|---|---|---|
| Plant type | 1 | 3.33 | 0.0679 |
| Treatment (Ag ppm) | 3 | 12.13 | 0.0069 |
| Plant type × Treatment (Ag ppm) | 3 | 8.68 | 0.0338 |

## Statistical analysis

To evaluate the effects of nanopriming on wild and domesticated chili plants, a logistic ANOVA and a survival analysis was used to analyze germination data. The model included the fixed effects of plant type (wild *vs.* domesticated), AgNP treatment concentration, and their interaction (type × treatment). Growth and biochemical traits—including wet and dry biomass, root and shoot length, total phenolic content, and chlorophyll content—were analyzed using two-way analysis of variance (two-way ANOVA). The fixed effects included plant type, AgNP concentration, and their interaction. Additionally, variation in mitotic phase frequencies was assessed using two-way ANOVA with plant type and AgNP exposure as fixed effects, along with their interaction. All statistical analyses were performed using JMP version 10 (SAS Institute Inc., Irvine, CA, USA).

## RESULTS

### Germination

Exposure to silver nanoparticles significantly affected total seed germination, with marked differences observed between wild and domesticated genotypes (Table 1; Fig. 1). In the control group, germination reached 77%, whereas treatment with 50 ppm AgNPs increased germination to 82%, 100 ppm resulted in 90%, and 250 ppm resulted in 91%. A significant plant type × treatment interaction was detected by ANOVA, indicating that the effect of AgNP concentration differed between wild and domesticated chili plants. Only the wild variety exhibited a dose-dependent response to AgNPs, with higher germination percentages observed at 100 ppm and 250 ppm compared to its control (Fig. 1). In contrast, the domesticated variety (Serrano) did not show significant differences in germination among AgNP treatments and the control. Germination rates for wild and domesticated chilies were 86% and 83.5%, respectively, and did not differ significantly ($\chi^2 = 3.33$, $P = 0.06$). However, within the wild variety, a significant difference was found between the control group and the 250 ppm treatment (Table S1). No significant differences in germination rate were observed among treatments for the domesticate (Table S1).

### Growth indicators

At 28 days after germination, significant differences between wild and domesticated varieties were observed only in shoot length, total plant length, total wet biomass (root and shoot combined), dry shoot weight, and total dry biomass (Table S2). No significant effects of AgNP treatment or treatment × variety interactions were detected at this stage.

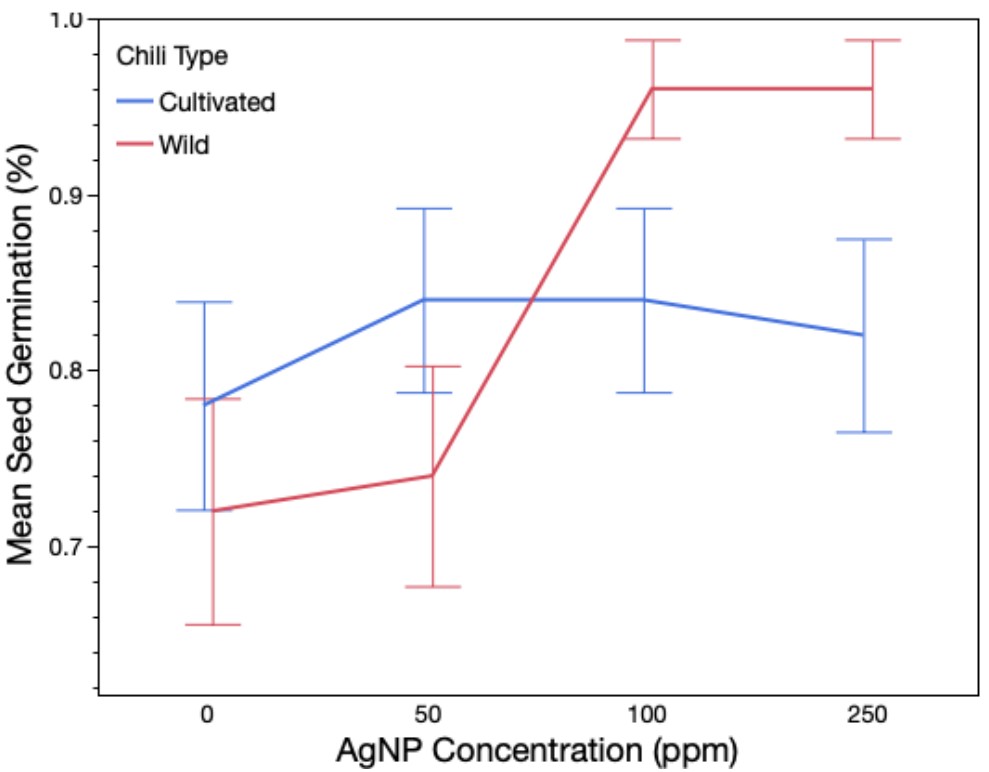

**Figure 1 Total germination of wild and domesticated *Capsicum annuum* plants exposed to three different doses of Ag nanoparticles.** The figure shows mean germination and confidence intervals obtained from a logistic ANOVA. Red line refers to wild genotype (Chiltepin); blue line refers to domesticated genotype (Serrano).

By 42 days, a significant treatment effect was detected only for total wet biomass ($p = 0.04$; Table S3), although this effect was independent of plant variety. A significant treatment × variety interaction was found only for total plant length ($p = 0.05$; Table S3; Fig. S1), indicating that the effect of AgNP concentration on total length varied between wild and domesticated plants.

## Total polyphenol content

Total polyphenol levels showed no significant differences between plant varieties or AgNP treatments at 28 and 42 days after germination (Table S4). No significant plant type × treatment interactions were detected. However, in the domesticated Serrano variety, polyphenol content tended to increase at the higher AgNP concentrations (100 and 250 ppm; Fig. S2). In contrast, the wild variety exhibited irregular and inconsistent patterns of polyphenol accumulation across treatments and time points. The lack of a consistent trend may reflect underlying differences in metabolic activity at various developmental stages.

## Chlorophyll content

Chlorophyll content differed significantly between wild and domesticated varieties at both 28 and 42 days after germination (Table S5). At 42 days, the effect of AgNP treatment was

variety-dependent: increasing nanoparticle concentrations led to a reduction in chlorophyll content in the domesticated variety, while the wild variety remained largely unaffected (Fig. S3).

## Cell division indicators: phytotoxic and genotoxic evaluation

Measurements of stem and root length during the first 72 h of AgNP exposure showed significant differences between the wild and domesticated varieties ($p < 0.0001$; Table S6), but no significant effects of AgNP treatments on shoot or root elongation were observed. Two-way ANOVA of leaf area at 48 and 72 h indicated significant variety × treatment interactions (Table S6); however, subsequent Tukey tests identified significant differences only between varieties, suggesting that AgNP treatments did not induce phytotoxic effects (Table S7).

Genotoxic evaluation showed no significant effects of AgNP treatment on the mitotic index or chromosomal aberrations in root cells for either variety (Table S8). Interestingly, in the wild chili, AgNP exposure increased the number of dividing cells compared to the control, while in the domesticated variety, a reduction in all mitotic phases was observed under AgNP treatment. This contrasting response may reflect underlying genetic differences between wild and domesticated genotypes.

## DISCUSSION

The use of silver nanoparticles (AgNPs) in seed nanopriming has gained increasing attention due to their potential to enhance germination and early seedling growth across various plant species. However, most studies have focused on single genotypes (*Mahakham et al., 2017*; *Acharya et al., 2020*; *Almutairi & Alharbi, 2015*; *Imtiaz et al., 2023*), with limited attention to understand whether wild and domesticated plants responses to nanomaterials can differ as a result of domestication history. In this study, we focused in investigating the effects of AgNPs on the germination, early growth, and physiology of wild and domesticated varieties of *Capsicum annuum*. Our results revealed that higher concentrations of AgNPs improved germination rates in wild seeds, while no significant effects were detected in the domesticated variety. Interestingly, we observed no detrimental changes in plant growth, and primary or secondary metabolism, such as chlorophyll and polyphenol leaf content. Finally, no evidence of cytotoxic or genotoxic effects in meristematic tissues was detected. Altogether our results demonstrate nanopriming with AgNPs may improve germination without adversely affecting the early development of these *C. annuum* varieties. These findings raise important questions about the effects of differential responses to nanopriming, particularly in the context of domestication, and remark considering evolutionary history as a factor on the application of nanopriming in crops and wild relatives.

## Variability in response to AgNPs between wild and domesticated varieties

The effects of AgNPs on plants have shown considerable variability across studies, depending on species identity and nanoparticle concentration (*Thongmak et al., 2022*;

*Mays et al., 2024*). The influence of natural history and domestication status has been poorly considered, despite growing evidence that domestication profoundly alters plant genotypes, phenotypes, and their functional responses (*Milla et al., 2015*; *Serrano-Mejía et al., 2022*). To our knowledge, this is the first study to explicitly compare the effects of AgNP nanopriming between a domesticated plant and its wild relative. Our results using wild and domesticated *Capsicum annuum* suggest that seed biological traits are key determinants of plant response to AgNP treatments. In the wild genotype, nanopriming with higher concentrations (100 ppm and 250 ppm) significantly increased germination rates, indicating a dose-dependent benefit. In contrast, the domesticated variety exhibited no significant response to any of the tested concentrations. These findings support the hypothesis that domestication-related changes in seed biology modulate the effectiveness of nanopriming.

The contrasting germination responses observed between wild and domesticated *C. annuum* varieties may be partially explained by physiological mechanisms influenced by nanopriming. In wild seeds, nanopriming may promote germination by enhancing physiological processes such as water imbibition and hormonal activation. The absorption of AgNPs by seeds has been shown to induce physiological responses related to the activation of phytohormones involved in growth and dormancy release (*Méndez-Argüello et al., 2016*), as well as the expression of genes related to cell proliferation (*Qian et al., 2013*). For instance, nanopriming can promote the activity of $\alpha$-amylase, a critical enzyme for breaking down starch reserves during germination (*Mahakham et al., 2017*). In wild chilies, low germination rates have been attributed to limited gibberellic acid (GA$_3$) availability under natural conditions. Treatments with exogenous GA$_3$ at various concentrations have led to increased germination in wild seeds (*Hernández-Verdugo, Oyama & Vázquez-Yanes, 2001*; *Cano-Vazquez et al., 2015*), suggesting that hormonal limitation plays a relevant role. Therefore, silver nanoparticles may facilitate the hormonal signals needed to trigger germination, offering a possible explanation for the enhanced germination observed in wild seeds.

An important question that arises is why the domesticated variety did not respond to AgNP nanopriming. Two potential explanations may account for this lack of response. One possibility is that the domestication process has reduced the sensitivity of *Capsicum annuum* seeds to nanoparticles. However, this explanation seems unlikely, as previous studies have reported variation in germination and seedling growth among domesticated *C. annuum* varieties in response to different types of AgNPs (*Yuan et al., 2018*; *Sánchez-Pérez et al., 2023*). A more plausible explanation is that domesticated seeds have evolved a more efficient physiological response to hydropriming, reducing their reliance on environmental signals to activate gibberellic acid or cytokinins required for germination. Therefore, while AgNPs may enhance germination in wild chilies, their application may not further improve germination in domesticated varieties that already exhibit high responsiveness to hydropriming.

## Evaluation of the cytotoxic and genotoxic impact of AgNPs

An important objective of this study was to determine whether exposure to AgNPs induces cytotoxic or genotoxic effects in the meristematic tissues of wild and domesticated *Capsicum annuum* varieties. In our study, primed and non-primed seedlings exhibited similar growth rates, reached comparable sizes, and accumulated similar levels of foliar chlorophyll and phenolic compounds 28 days after germination (Table S1). Moreover, further assays showed no negative effects of AgNPs on apical or root meristems, supporting the conclusion that AgNPs did not exert phytotoxic effects on plant growth. These findings are consistent with previous studies reporting no phytotoxicity of silver nanoparticles at low to moderate concentrations (*Budhani et al., 2019*). For example, in *Vanilla planifolia*, concentrations of 25 and 50 mg/L promoted growth without causing significant genotoxic effects (*Spinoso-Castillo et al., 2017*; *Bello-Bello et al., 2018*). In *Allium cepa*, concentrations up to $100\,\mu g/mL$ of AgNP solution promoted growth without causing genotoxic or cytotoxic damage (*Casillas-Figueroa et al., 2020*). However, studies in other plant systems have associated chromosomal abnormalities to prolonged exposure to higher concentrations (*Kumari, Mukherjee & Chandrasekaran, 2009*). We observed no phytotoxic effects even at higher concentrations, suggesting that both wild and domesticated *Capsicum annuum* can tolerate elevated levels of silver nanoparticles, supporting their potential as a safe tool for enhancing germination and growth in this plant species.

One limitation of this study is the absence of PVP and ionic silver treatments to assess their potential effects on germination and growth. The experiment was not designed to isolate the effects of PVP, as previous studies have demonstrated that the PVP coating of the AgNPs used does not induce cytotoxic or genotoxic effects at the applied concentrations (*Casillas-Figueroa et al., 2020*; *Bello-Bello et al., 2018*). Moreover, other studies have reported that the observed biological effects are primarily associated with the nanoparticles themselves, rather than the coating agent or released silver ions (*Cvjetko et al., 2017*). Therefore, the effects observed in this study are most likely attributable to the nanoparticles.

## CONCLUSIONS

Silver nanoparticles (AgNPs) represent a promising tool to address challenges in germination and early seedling development, particularly in plant varieties with low germination rates. This study demonstrates that AgNP priming improves germination in wild *Capsicum annuum* seeds and provides evidence of the treatment's safety, as no phytotoxic or genotoxic effects were observed. Although domesticated seeds showed no significant response, the findings remark the potential of AgNPs to support germination in wild varieties without compromising early growth or metabolic function. However, plant responses to nanoparticle exposure vary substantially depending on the genetic background and the physiological traits of plants, and particle characteristics (*e.g.*, type, size, concentration, and surface chemistry), reflecting the complexity of these interactions (*Thuesombat et al., 2014*; *Arruda et al., 2015*; *Thongmak et al., 2022*). To produce more generalizable conclusions, future experiments should test a specific nanoparticle type

across multiple species and crops, or multiple nanoparticle types within a single model species. For example, additional studies in Capsicum should include multiple accessions—encompassing wild and domesticated varieties or species—to determine whether AgNP effects extend to other Capsicum cultivars.

## ACKNOWLEDGEMENTS

We thank Dr. Olivia Torres for her valuable guidance on the preparation of tissue samples for the genotoxicity analysis. We are also grateful to the members of the Genetics, Ecology, and Evolution Laboratory (GGE Lab) for their assistance during the experiments.

### Funding

This work was supported by UABC under the Convocatoria Interna de Proyectos de Investigación 2023–2025. The equipment used was funded through CONACYT grant INFRA-2014-01-226239 awarded to Rafael Bello-Bedoy. Ramón Carrillo supported this study by the "Convocatoria Interna de Proyectos de Investigación 2024_2025_UABC (# 252)" CONACYT grant CB 2015-1 #255631. The funders had no role in study design, data collection and analysis, decision to publish, or preparation of the manuscript.

### Grant Disclosures

The following grant information was disclosed by the authors:
UABC.
CONACYT: INFRA-2014-01-226239.
Convocatoria Interna de Proyectos de Investigación 2024_2025_UABC (#252): CONACYT grant CB 2015-1 #255631.

### Competing Interests

The authors declare there are no competing interests.

### Author Contributions

- Berenice Cortes Espinoza conceived and designed the experiments, performed the experiments, analyzed the data, prepared figures and/or tables, authored or reviewed drafts of the article, and approved the final draft.
- Alejandro Sánchez-González conceived and designed the experiments, authored or reviewed drafts of the article, and approved the final draft.
- María Evarista Arellano-García conceived and designed the experiments, authored or reviewed drafts of the article, and approved the final draft.
- Rafael Bello-Bedoy conceived and designed the experiments, performed the experiments, analyzed the data, prepared figures and/or tables, authored or reviewed drafts of the article, and approved the final draft.

### Data Availability

The raw measurements are available in the Supplementary Files.

## Supplemental Information

Supplemental information for this article can be found online at http://dx.doi.org/10.7717/peerj.19974#supplemental-information.

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
