# Peer review of "Beneficial dose-dependent effects of Ag nanoparticles on germination do not compromise growth and metabolic profiles of Capsicum annuum seedlings"

_PeerJ, doi:10.7717/peerj.19974_

## Round 0.1 · original submission · Major Revisions

Dear Dr. Bello-Bedoy, I ask you to review your article carefully and consider the reviewers' suggestions. The study has shortcomings, but they can be easily addressed by careful discussion and citation of relevant studies.

**Language Note:** The review process has identified that the English language must be improved. PeerJ can provide language editing services - please contact us at [email protected] for pricing (be sure to provide your manuscript number and title). Alternatively, you should make your own arrangements to improve the language quality and provide details in your response letter. – PeerJ Staff

Reviewer 1 ·

Basic reporting

The text needs to be completely revised. It contains too many errors, for example:
- Line 55: “Seed priming using nanoparticles (i.e., nano priming)”.
Correction: nanopriming
- Line 57: “The application of 57 nano primming can induce”.
- Correction: nanopriming
- Line 60: “However, the effects of nano priming”
- Correction: nanopriming
- Line 67: “Nanoprimming effects with Silver nanoparticles”
- Correction: nanopriming, silver
- Line 71: “in a diploid cultivar exposed nanoprimming”
- Correction: nanopriming
- Line 72: “with silver particles, but its effects”
- Correction: nanoparticles
- Line 75: “effects of AgNP nanoprimming”
- Correction: nanopriming
- You should pay attention to the words "nano priming, nanopriming, nano primming, nanoprimming, nanoparticle (the "s" is missing), particle (it should be nanoparticles).

- The abbreviations "AgNP" and "AgNPs" are used interchangeably. Their use should be consistent. "AgNps"

- Lines 131-132: “4.16 (207.9mg/50 seeds) and wild chili seeds had a mean weight of 2.6mg (132.7 mg/50 seeds)”

Correction: 4.16 mg, 207.9 mg, 2.6 mg (put a space between the value and the unit of measurement)
- Line 137: “Each treatment had four replicates”
The suggestion: “Each treatment was replicated four times”

- Line 169: “o the nearest 0.0001 gr”.
Correction: 0.0001 g.

- Lines 143-145: “Day to germination was registered since the day there were treated. The number of germinated seeds was counted every day and along 14 days of germination where most seeds had germinated”.

Suggestion: Germination was recorded from the day of treatment. The number of germinated seeds was counted daily over the 14 days of germination, when most seeds had germinated.

- Line 179: Subscript in the formula for sodium carbonate
- Line 186: Superscript in square centimeter
- Line 304: “(100ppm and 250 ppm)”.

The entire text needs to be reviewed, as it includes a value without a space followed by its unit of measurement, and in others, there is a space between the value and the unit.

- Line 222: “concentration were measured by means of a two wan analysis of variance” – two-way?

Experimental design

- The study uses commercial silver nanoparticles. At least one TEM or SEM image of the reagent batch used must be included, or at least the nanoparticle size and polydispersity must be specified. Small changes in shape or size can cause comparable changes.

- When using commercial nanoparticles recovered with PVP, there is no comparison as to whether this PVP acted in any way, or if there is a report specifying that it does not affect in a similar study at the concentration at which the AgNPs were used.

- A simple question: Why were sugars not studied as part of metabolites, or any other than just polyphenols?

- Line 146: “Total germination percentage (%) and germination rate. The germination percentage (G%)”.
% and G% are not the same?

- Line 175: “1 mL of hydro-methanolic solution (methanol 100%) for 24 hours in the dark”.
Observation: It is confusing to use the term "hydro-methanolic solution" and specify that it is 100% methanol. Please correct that.

- Lines 176-182: Was the protocol carried out as described in the reference “Ainsworth & Gillespie, 2007”? If the protocol was followed, please specify it directly, since you do not mention any concentration of the reagents.

Validity of the findings

- Was the potential effect of silver as a free ion versus a nanoparticle monitored? This is important to understand whether the response is due to the nanoparticle effect or simply due to the silver ionic effect.
https://doi.org/10.3390/ijms21103441
If experimental data are not available, this point should be fully discussed with appropriate references.

- Lines 327-338 It's a good discussion, but it reads more like a conclusion.

- Lines 336-338: There are references cited (Song & He, 2021; Thongmak et al., 2022), raising doubts about whether the final lines are taken from these authors or are actually part of this discussion.
"Further research is needed to understand how domestication has shaped
germination mechanisms and their relationship to nanoprimming technology (Song & He, 2021; 338 Thongmak et al., 2022).

Additional comments

- Lines 382-387: "Future research should focus on understanding the physiological and molecular mechanisms underlying the interaction between nanoparticles and seeds (Mahakham et al., 2017)"
This idea is mentioned a couple of times in the discussion. The question is: What did the authors do to carry out these new studies that address the questions raised by others?

- The terminology alternates between "wild vs. cultivated" and "wild vs. domesticated." This can cause confusion. It's advisable to maintain consistent terminology throughout the manuscript.
Consider that domestication is an evolutionary process in which the genetic characteristics of a wild species are modified to adapt it to human needs. Cultivation, on the other hand, refers to the management of plants, whether domesticated or wild, for their growth and production.

·

Basic reporting

no comment

Experimental design

no comment

Validity of the findings

no comment

Additional comments

Review is attached.

---

## Round 0.2 · accepted · Accept

Dear Dr. Bello-Bedoy, I congratulate you on the acceptance of this article for publication and wish you further success in your scientific research.

The Section Editor recommends modifying the title to "..growth and metabolic profiles.."

Reviewer 1 ·

Basic reporting

No comment

Experimental design

No comment

Validity of the findings

No comment

Additional comments

The authors have made the necessary changes. I don't have any more comments. The article may be accepted for publication.